# Factors Affecting Antiretroviral Therapy Adherence among HIV-Positive Pregnant Women in Greece: An Exploratory Study

**DOI:** 10.3390/healthcare10040654

**Published:** 2022-03-31

**Authors:** Georgia Pontiki, Antigoni Sarantaki, Petros Nikolaidis, Aikaterini Lykeridou

**Affiliations:** 1“Elena Venizelou” Hospital, 115 21 Athens, Greece; gpontiki@hotmail.com; 2Department of Midwifery, Faculty of Health & Care Sciences, University of West Attica, Egaleo, 122 43 Athens, Greece; pnikol@uniwa.gr (P.N.); klyker@uniwa.gr (A.L.)

**Keywords:** HIV, AIDS, pregnancy, antiretroviral therapy, compliance, adherence, knowledge, attitudes

## Abstract

The human immunodeficiency virus (HIV) is a major public health problem globally. Each year, approximately 1.4 million women living with HIV get pregnant. This contemporary descriptive study investigates the degree of compliance of HIV-positive women-patients undergoing antiretroviral therapy (ART) during pregnancy. A sample of 200 treated HIV-positive pregnant women (mean age, 32.9 years; Greek nationality, 67.5%; poor educational level, 28.5%) was selected. The data collection occurred in three acquired immunodeficiency syndrome (AIDS) reference centers in Athens, Greece, from November 2019 to September 2021. Patients’ median knowledge score was 50% (IQR: 38.9–61.1%), and their median attitude score was 4.2 (IQR: 3.6–4.4); 13.0% of participants did comply with ART treatment. Specifically, 7.0% of them failed to take their treatment twice when asked about their activities over the preceding 7 days, and 3.0% skipped it three times. Women of Greek nationality had significantly higher compliance with treatment (*p* < 0.001). Additionally, a higher compared to lower education level was significantly associated with greater compliance (*p* = 0.001), while women with a low level of social support had significantly lower compliance. Participants who had complied with ART had significantly higher knowledge and attitude scores (*p* = 0.027). Patient characteristics determine compliance with ART in HIV-positive pregnant women in Greece, while the availability and quality of health system services may modulate this relationship.

## 1. Introduction

Human immunodeficiency virus (HIV) is one of the major global public health problems [1]. Each year, about 1.4 million women with HIV get pregnant, and these pregnancies lead to approximately 220,000 new infections in infants and children [2]. Thus, it is crucial to take care of HIV-positive pregnant women, but it is also essential to reduce the vertical transmission of HIV from mother to child [3].

Antiretroviral therapy (ART) has improved the quality of life of HIV-positive individuals with or without symptoms. A reduction in HIV morbidity and mortality has been recognized in countries where ART is widely available [4]. However, ART requires a lifelong implementation and strict adherence to the medication [5]. In order to achieve optimal results from the systemic administration of ART, there must be a high level of compliance with the treatment (at least 95%). Nevertheless, there are many barriers to compliance worldwide [4], at least partly related to social, cultural, and demographic patient characteristics, facilitated approaches to health system providers, and ART-related adverse events. Thus, it is important to identify the factors that hinder compliance in different countries, and to develop strategies to ensure long-term compliance [6]. In particular, pregnancy status may considerably influence compliance with ART, since reduced compliance can be enhanced by women’s beliefs of potential harm to the developing fetus. This attitude may impact maternal health and increase the rates of vertical HIV transmission, despite guideline recommendations to triple-dose ART for HIV-positive women during pregnancy and after childbirth [7]. Previous studies [1,3,8] and a systematic review, including cohorts mainly from developing countries [9], have examined the adherence to ART during both pregnancy and the postpartum period. Although child-care requirements can significantly affect the self-care behaviors of women during the postpartum period, additional barriers, including nationality and uninterrupted ART use, have been reviewed elsewhere [9].

Many HIV-positive women of a nationality other than Greek are getting pregnant in Greece. Moreover, no previous effort to determine the ART compliance rate in HIV-positive pregnant women has been undertaken. Therefore, we hypothesize that the compliance rate with ART in Greece might be lower compared to developing countries. However, important confounders, such as nationality and other individual or social characteristics of the studied HIV-positive pregnant cohort, should be considered when interpreting our findings.

## 2. Materials and Methods

### 2.1. Research Plan and Sample

The present study is a descriptive correlation study with synchronous planning to find factors related to compliance with ART. The study sample consisted of 200 HIV-positive pregnant women from three AIDS reference centers in Athens, Greece. The collection of questionnaires took place from November 2019 to September 2021. These individuals were selected by convenience sampling (non-random sampling) of all HIV-positive pregnant women.

### 2.2. Research Tool

For the needs of the present research study, a combination of weighted questionnaires was used, which have been used in studies abroad. More specifically, the questionnaires used were a social support questionnaire (“Oslo 3-Item Social Support (OSS-3) scale”), a depression scale (“PHQ-9”), a general knowledge questionnaire (“HIV Knowledge Questionnaire (HIV-KQ-18)”) [10], which is an attitude questionnaire compiled by the researcher, and the general questionnaire on compliance with ART (“Questionnaire on taking Antiretroviral Medication”) [11]. The licenses for using the questionnaires were obtained after contacting their authors. Afterward, the questionnaires’ translation, intercultural adaptation, and weighting were performed.

### 2.3. Criteria for Sample Selection

The study included all HIV-positive women who were willing to participate and came to the selected disease reference points during the study period. They had also started ART and spoke the Greek language sufficiently, or an interpreter/mediator was used.

### 2.4. Ethics

Permission was requested and granted for study conduction from the Hospital Ethics Committees of the three AIDS reference centers. All collected data were processed anonymously.

### 2.5. Outcome Definition

We defined low compliance with ART treatment whenever ART treatment was omitted at least one time during the last week. Conversely, high compliance included women who received all ART pills during the same period.

### 2.6. Data Analysis

Quantitative variables were expressed as means and standard deviations (SD), while qualitative variables were expressed as absolute numbers, relative frequencies, and interquartile ranges (IQR). In order to compare proportions, the Chi-square and Fisher’s exact tests were used. In addition, Student’s *t*-tests or Mann–Whitney tests were computed to compare quantitative variables between women who were compliant to treatment and those who were not. Multiple logistic regression analysis with a stepwise approach (*p*-value for entry 0.05, *p*-value for removal 0.10) was used, setting as the dependent variable the compliance with treatment. The logistic regression analyses computed adjusted odds ratios (ORs) with 95% confidence intervals (CIs). All reported *p* values were two-tailed. Statistical significance was set at *p* < 0.05, and analyses were conducted using the SPSS statistical software (version 22.0).

## 3. Results

The sample consisted of 200 HIV-positive pregnant women, with a mean age of 32.9 ± 5.1 years (Table 1). Most of the patients were Greek (67.5%), but the high school graduates made up 40.5% and the employed 58.5% of the overall cohort. Almost half of the patients (46.9%) were married, 76.5% lived with their partner/children and/or other family members, and 32.5% had a monthly income of more than EUR 750. Moreover, 3.0% of the sample consumed alcohol, and 13.0% used addictive substances. Additionally, 5.5% of the sample were diagnosed with a chronic systemic disease, and 1.5% had a chronic psychological disease. The mean gestational age was 37.4 ± 0.8 weeks, and the mean PHD-9 score was 7.7 ± 4.6. Overall, 36.5% of the participants had moderate social support. Patients’ median knowledge score was 50% (IQR, 38.9–61.1%), and their median attitude score was 4.2 (IQR, 3.6–4.4).

The activities of the week prior to study participation and information regarding patient compliance to treatment are presented in Table 2. The most frequent activity of the participants was being visited by friends or family members (89.5%), followed by participation in a meeting and visits to friends or family members, being 67.5% and 66.5%, respectively. A total of 13.0% of participants did not comply with treatment. Specifically, 7.0% of them failed to take their treatment twice when asked about their activities in the preceding 7 days, and 3.0% of them skipped it three times.

Table 3 reports univariate associations of treatment compliance with different socio-demographic characteristics. Greek women had significantly higher compliance with treatment than women of other nationalities. A higher educational level was significantly associated with greater compliance than a lower level. The compliance of participants who lived with their spouse/partner and/or children and other family members was significantly higher than those who did not. In contrast, those who adhered to treatment were younger than their non-compliant counterparts. Moreover, women with a low level of social support had significantly lower compliance than women with a higher level of social support. Significantly higher treatment compliance rates were found in participants who went out for entertainment, those who went to a bar, and those who had attended a meeting in the last week, than in less socialized women. Finally, participants who had complied with treatment had significantly higher knowledge and attitude scores than non-compliant women.

A stepwise logistic regression analysis (Table 4), after adjustment for confounders (i.e., age, gestational age, monthly income, living status, family status, alcohol consumption, and addictive drugs), revealed the significant determinants of compliance with ART. Indeed, women with a nationality other than Greek had an 80% lower probability of compliance with treatment when compared to Greek women. At the same time, higher educational levels and greater knowledge scores were associated with a 2.32- and 2.24-times higher probability of compliance with treatment, respectively.

Compliance with ART depends on complex dynamic behavior influenced by various factors related to the individual, the health system, and socioeconomic environments.

The compliance of pregnant HIV-positive women with ART becomes even more difficult when addiction, depression, and ART coexist [12]. More than 10% of the participants in our study used addictive substances. However, this rate is not divergent from the mean rate (i.e., 12.1%) of women from the European Union who are younger than 34 and who have used addictive substances in the last year [13]. The rate of women who used alcohol in our study was slightly lower than 10%. However, our study’s percentage of alcohol use was much lower than that of Washio et al. [14]. In this latter study, although almost half of the participants reported alcohol use in the previous month, after the behavioral intervention, they improved adherence to ART and reduced alcohol use.

Numerous studies have shown that higher compliance treatment rates were achieved when health education was combined with strategies for behavior modification and emotional support of patients [15]. However, in our study, 36.5% of the participants had indifferent social support. Positive emotional and social support, especially from partners, was associated with increased compliance with treatment and fewer signs of depression [16,17,18]. Conversely, when the level of emotional support was low, women were not satisfied with the social support they received, including the practices of health professionals [19].

Although the majority of the participants complied with treatment, 13% did not. In many other studies, the compliance rates were similar when the aggregate percentage of women with satisfactory compliance was high prenatally. It was higher than that after childbirth [12,20,21]. However, several studies showed that the rate of non-compliance with treatment was higher, reaching over 20% [22,23]. The non-compliance rates were also high in the study by Oginni et al. [24]. The majority of participants (53%) did not receive medication at baseline, and the definition of compliance with ART maintenance or initiation was less strict than in our cohort. More specifically, in the current study, 7% of pregnant women failed twice to receive treatment when asked about their activities in the preceding 7 days, and 3% had missed it three times. In other studies, the omission of one dose was not a criterion for including women in the non-compliant group [1,22,25]. Indeed, skipping more than one dose of ART was higher in the study by Bailey et al. [26] and in the study by Nutor et al. [2] (35% and 37.8%, respectively), a condition partially explained by the younger age. Overall, the findings that emerged from the correlation of treatment compliance with demographics and other study characteristics, and compared with other studies worldwide, were remarkably similar [27,28].

In previous studies, educational level as a determinant of compliance was rather conflicting. Patients with a higher level of education adhere to the treatment thoroughly, attributing their commitment to better information and processing of information [29]. By contrast, the focus on treatment was higher in patients with a lower educational level due to greater trust in health professionals and the fact that they see physicians as authorities [30]. In the present study, the higher the participants’ level of education, the higher the compliance rate was. On the other hand, in many studies, low educational attainment is significantly associated with the non-compliance of pregnant women [1,31].

Cultural factors influence compliance with treatment due to lower socioeconomic status and language barriers [32,33]. The available evidence points to a link between cultural factors and inequalities in the treatment of HIV, which are correlated not only with the socioeconomic characteristics of specific cultural groups (such as poverty and low educational level) but also with mental health [34,35]. Indeed, Greek women had significantly higher treatment compliance rates than women of other nationalities living in Greece in the present study. This finding is partially explained by the barriers encountered by immigrant women in health-care settings, or cultural beliefs from countries of origin (including HIV denial). Likewise, in the study by Mellins et al. [3], one of the factors associated with non-compliance, especially in the postpartum period, was the ethnicity of HIV-positive women. Studies have proved that unemployed women of color of low educational and socioeconomic status show lower compliance with ART. In contrast, women have been associated with developing resistance to ART, which is a sign of low compliance [36,37,38].

Another key indicator of compliance with treatment is health education [39]. Relevant studies have demonstrated a link between health education, self-efficacy, and medication compliance [40]. In addition, numerous studies have proved that the best results in compliance with treatment are achieved when health education is combined with behavioral modification strategies and the parallel emotional support of patients [15].

In our study, the average amount of knowledge was limited (50%), and the average attitude was satisfactory (4.2. (IQR: 3.6–4.4)). The knowledge percentage was quite low, similar to the study of Nutor et al. [41], where participants had less knowledge about HIV transmission. On the other hand, in many studies, most pregnant women had a basic understanding of the mother-to-child transmission of HIV during pregnancy [20,42,43]. However, the overall attitude score in the present study was high. Positive attitudes toward ART are related to promoting maternal health, preventing vertical transmission [44], accepting HIV, and seeking counseling to help women with HIV cope with their condition to reduce their negative emotions [45].

Finally, participants who had complied with treatment had significantly higher knowledge and attitude scores. In the systematic review by Omonaiye et al. [46], knowledge of HIV status and information about the disease, either before or during pregnancy, was significantly associated with medication compliance. Women who knew about their condition and acquired general knowledge about the disease before pregnancy showed better compliance than their lower-knowledge counterparts. Nevertheless, women who discovered the infection during pregnancy did not follow their treatment properly.

We acknowledge several limitations in our study. First, the coronavirus pandemic might be considered an emerging barrier to ART compliance due to (1) physician–patient distancing, (2) the reduction in regular medical visits, (3) the instauration of a non-healthy lifestyle, including social distancing and unhealthy behaviors, and (4) the reinforcement of inadequate prescription refill and adherence to medications. Second, our findings cannot be extended to cohorts from other geographic regions or countries. Third, our study design cannot suggest a cause–effect relationship between nationality, knowledge score, or educational level and ART compliance in pregnancy.

## 4. Conclusions

Compliance with ART in HIV-positive pregnant women in Greece is 13%, a rate almost 50% lower than that observed in developing countries. In addition, women of Greek nationality, compared to women of non-Greek nationality living in Greece, have better compliance rates to ART. Finally, a better compliance to ART is associated with a higher educational level or knowledge score.

Health professionals need to work with pregnant women and their “significant others” from the supportive environment in order to identify barriers and reduce them as much as possible, to enhance compliance with the treatment plan. Therefore, health professionals should develop relationships based on mutual trust and respect for service users’ needs, and promptly recognize the need to support individuals who comply with the proposed ART schemes.

## Figures and Tables

**Table 1 healthcare-10-00654-t001:** Study sample characteristics.

	N (%)
Age, mean (SD)	32.9 (5.1)
Gestational age (weeks), mean (SD)	37.4 (0.8)
Nationality	
Greek	135 (67.5)
Other	65 (32.5)
Educational level	
None/Primary school	57 (28.5)
Secondary school	50 (25.0)
High school	81 (40.5)
University	12 (6.0)
Employed	117 (58.5)
Monthly income	
EUR 0	51 (25.5)
EUR 1–500	49 (24.5)
EUR 500–750	35 (17.5)
EUR > 750	65 (32.5)
Living status	
Alone	6 (3.0)
Partner and/or Children and/or other family members	153 (76.5)
Children and/or other family members	25 (12.5)
Other family members only	16 (8.0)
Family status	
Married	90 (46.9)
Divorced	17 (8.9)
Unmarried	85 (44.3)
Alcohol consumption	6 (3.0)
Substance use	26 (13.0)
Chronic somatic disease	11 (5.5)
Chronic psychological disease	3 (1.5)
Use of alternative treatments	3 (1.5)
Attend childbirth preparation courses by maids	20 (10.0)
In-hospital specialized programs for following pregnancies of HIV patients	14 (7.0)
Collaboration with maids for resolving questions	12 (6.0)
Having a regular contact with a maid contributes in better compliance to treatment	191 (95.5)
Health services should be more organized in pregnant women with HIV	200 (100.0)
Social support level	
Low	68 (34.0)
Moderate	73 (36.5)
High	59 (29.5)
PHQ-9 score, mean (SD)	7.7 (4.6)
Knowledge score (%), median (IQR)	50.0 (38.9–61.1)
Attitude score, median (IQR)	4.2 (3.6–4.4)

**Table 2 healthcare-10-00654-t002:** Patients’ activity in the preceding week and compliance to treatment.

	N (%)
During last week, you:	
Went out for entertainment	118 (59.0)
Went to a restaurant	80 (40.0)
Went to a bar	27 (13.5)
Went to a party	9 (4.5)
Slept in another house	53 (26.5)
Participated in a meeting	133 (66.5)
Visited friends or family	135 (67.5)
Had friends or family over for a visit	179 (89.5)
Did any of the above, stopped you from taking your medication	26 (13.0)
Compliance to treatment	174 (87.0)
During last week, how many times did you skip taking one or more of your antiretroic medicine?	
0	174 (87.0)
1	6 (3.0)
2	14 (7.0)
3	6 (3.0)
This corresponds in how many antiretroic pills?	
2	12 (46.2)
4	8 (30.8)
6	6 (23.1)

**Table 3 healthcare-10-00654-t003:** Univariate analysis for the association of patients’ characteristics with compliance to treatment.

		Compliance to Treatment		
		No	Yes	
		N (%)	N (%)	*p*
Age, mean (SD)		34.8 (3.9)	31.8 (5.3)	0.006 +
Nationality	Greek	9 (6.7)	126 (93.3)	<0.001 ‡
	Other	17 (26.2)	48 (73.8)	
Educational level	None/Primary school	15 (26.3)	42 (73.7)	0.004 ‡
	Secondary school	5 (10.0)	45 (90.0)	
	High school	6 (7.4)	75 (92.6)	
	University	0 (0.0)	12 (100.0)	
Employed	No	12 (14.5)	71 (85.5)	0.606 ‡
	Yes	14 (12.0)	103 (88.0)	
Monthly income	EUR 0	9 (17.6)	42 (82.4)	0.411 ‡
	EUR 1–500	8 (16.3)	41 (83.7)	
	EUR 500–750	3 (8.6)	32 (91.4)	
	EUR > 750	6 (9.2)	59 (90.8)	
Living status	Alone	0 (0.0)	6 (100.0)	0.021 ‡
	Partner and/or Children and/or other family members	26 (17.0)	127 (83.0)	
	Children and/or other family members	0 (0.0)	25 (100.0)	
	Other family members only	0 (0.0)	16 (100.0)	
Substance use	No	23 (13.2)	151 (86.8)	1.000 ‡‡
	Yes	3 (11.5)	23 (88.5)	
Chronic systemic disease	No	23 (12.2)	166 (87.8)	0.158 ‡‡
	Yes	3 (27.3)	8 (72.7)	
Attend childbirth preparation courses by maids	No	23 (12.8)	157 (87.2)	0.729 ‡‡
	Yes	3 (15.0)	17 (85.0)	
In-hospital specialized programs for following pregnancies of HIV patients	No	26 (14.0)	160 (86.0)	0.223 ‡‡
	Yes	0 (0.0)	14 (100.0)	
Collaboration with maids for resolving questions	No	26 (13.8)	162 (86.2)	0.372 ‡‡
	Yes	0 (0.0)	12 (100.0)	
Social support level	Low	18 (26.4)	50 (73.5)	0.015 ‡
	Moderate	8 (11.0)	65 (89.0)	
	High	6 (10.2)	53 (89.8)	
During last week, you:				
Went out for entertainment	No	20 (24.4)	62 (75.6)	<0.001 ‡
	Yes	6 (5.1)	112 (94.9)	
Went to a restaurant	No	20 (16.7)	100 (83.3)	0.059 ‡
	Yes	6 (7.5)	74 (92.5)	
Went to a bar	No	26 (15.0)	147 (85.0)	0.029 ‡‡
	Yes	0 (0.0)	27 (100.0)	
Went to a party	No	26 (13.6)	165 (86.4)	0.609 ‡‡
	Yes	0 (0.0)	9 (100.0)	
Slept in another house	No	18 (12.2)	129 (87.8)	0.597 ‡
	Yes	8 (15.1)	45 (84.9)	
Participated in a meeting	No	15 (22.4)	52 (77.6)	0.005 ‡
	Yes	11 (8.3)	122 (91.7)	
Visited friends or family	No	12 (18.5)	53 (81.5)	0.111 ‡
	Yes	14 (10.4)	121 (89.6)	
Had friends or family over for a visit	No	0 (0.0)	21 (100.0)	0.082 ‡‡
	Yes	26 (14.5)	153 (85.5)	
Gestational age (weeks), mean (SD)		37.2 (1.0)	37.4 (0.8)	0.150 +
Knowledge score (%), median (IQR)		50 (38.9–50)	50 (38.9–66.7)	0.025 ++
Attitude score, median (IQR)		3.60 (3.60–4.20)	4.2 (3.6–4.6)	0.011 ++
PHQ-9 score, mean (SD)		7.83 (5.70)	7.65 (4.48)	0.865 +

+ Student’s *t*-test; ++ Mann–Whitney test; ‡ Pearson’s Chi-square test; ‡‡ Fisher’s exact test.

**Table 4 healthcare-10-00654-t004:** Stepwise multivariate logistic regression analysis exploring the association between ART compliance and relevant determinants after adjustment.

		OR +	95% CI ++	*p*
Nationality	Greek (reference)			
	Other	0.20	0.08–0.48	<0.001
Educational level		2.32	1.40–3.84	0.001
Knowledge score (%)		2.24	1.10–4.56	0.027

+ Odds Ratio; ++ 95% Confidence Interval.

## Data Availability

The data presented are included in this study; additional data may be provided by the corresponding author on request.

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
