# Peer review of "Factors Affecting Antiretroviral Therapy Adherence among HIV-Positive Pregnant Women in Greece: An Exploratory Study"

_healthcare, 2022, doi:10.3390/healthcare10040654_

Round 1
Reviewer 1 Report
I appreciate the authors’ revision of the manuscript. I would like them to consider some points listed below:
l.154 Please show the figure of “the mean rate” and its source.
l.201 It seems inappropriate, or lacking supporting data, to attribute lower compliance of emigrants to COVID-19 pandemic, because the study period includes pre-pandemic, and influence of the pandemic is not investigated.
l.242 Please add ”better” to “compliance” to show there is positive correlation between compliance and education/knowledge.
Reviewer 2 Report
The authors wrote "The compliance of participants who 128 lived with their spouse/partner and/or children and other family members was 129 significantly lower than the opposite." lines 128-130. The results in Table 3 show the opposite. I believe this is a typo in the Results section, rather than the data presented in Table 3, and must be corrected prior to publication.
Reviewer 3 Report
The authors have addressed the majority of my earlier concerns.
Author Response
Please see attachment.

This manuscript is a resubmission of an earlier submission. The following is a list of the peer review reports and author responses from that submission.
Round 1
Reviewer 1 Report
The authors investigated the possible factors affecting ART among HIV+ pregnant women in Greece, found that compliance to ART is complex, and concluded that professional healthcare providers should be supportive to them.
Although their research contains little novelty and speculation from it could be applied to women in Greece only, the manuscript includes enough comparison with previous reports published in other countries, so it would be interesting to readers of the international journal “Healthcare”, with some revision.
l.51 “Few studies” seems an overstatement; as you quote them, there have been previous reports on compliance to ART. Please refer to them.
l.55 What is a difference between HIV-positive “women” and “patients”?
l.55 “in Greece” seems necessary.
l.101 “From a chronic somatic disease suffered” seems an error.
l.135 Please show a percentage of substance addictions among all Greeks or non-pregnant women, if possible.
l.211 Authors does not answer their own question presented at the end of the introduction section, that is, they should talk about “the degree of compliance with ART in HIV-positive women and patients during pregnancy (in Greece)”.
l.190 Table 4 shows five times compliance with ART among Greeks compared to other nationalities, but it is not discussed enough. What is the reason? And is there any clue to better compliance?
l.194 The expression “women of colour” seems inappropriate; it sometimes contains a politicism or a racism.
Reviewer 2 Report
The authors have aimed to assess the compliance of pregnant women with HIV treatments during the course of their pregnancies, using quantitative and qualitative criteria. While this is a valuable analysis, I have these concerns in assessing this manuscript as it was submitted:
Major points:
- Where are Table 1, 2, 3 and 4? Impossible to review this paper without these tables that contain the data.
- The authors have adapted three questionnaires from others. There are no tables in this manuscript as submitted and there are no additional explanations about these questions in the Methods or within the Results section. These must be included in the manuscript.
Without the tables, and in the absence of the questionnaires that the authors have utilized, it is not possible to fairly assess the current manuscript.
Minor points:
- In the Ethics section of M&M (lines 80-82), specify the authorities from which the permissions were sought and obtained.
- Line 84, (SD) is abbreviation for standard deviation not mean. Correct/clarify the sentence.
- Line 101 is unclear.
- Most patients were Greek, but not high school graduates (40% is not most). Correct the sentence as it reads. Line 97. If it is a typo, correct it. The Table is missing, so impossible to know.
- For clarity, change the “activities of the last week” to the activities of the week prior to….
- “Line 114: Greek women had significantly higher percentages of compliance with treatment 114 (Table 3).” Compared to what? The table is not provided. But this should be mentioned in the text as well. For example, is this compared to global rates of compliance among pregnant women? Or, compared to other non-pregnant women with HIV? Compared to the general population? Etc. It is unclear what the comparison is. Again, no tables, and no questionnaires.
- “Also, a higher educational level was significantly associated with greater percentages of compliance. Line 115”. Same problem as in point 6, above.
Reviewer 3 Report
Authors described about factors affecting ART adherence among HIV-positive pregnant women in Greece which is quite important topic, especially for preventing mother to child of HIV transmission.
Authors concluded that compliance is determined by health professional and health systems. But in the discussion there were no explanation about the role of health professional. How is the health system in country? How this factor affect compliance?
Authors wrote the sentence of "Compliance with treatment is a complex behaviour that is determined by factors related to the patient, health professionals and health systems" twice in the abstract and conclusion. Please use different expression, do not repeating what you have already said.
The conclusion is not focused to ensure the whole topic. I suggest to move sentences (227-232) to discussion section